# IDetail: Fine-grained Identity Preservation in Prompt-based Image Relighting

## Abstract

Diffusion-based methods are widely used for image-to-image translation tasks such as object addition/removal, colorization, and prompt-based editing. In personalized editing applications, accurately preserving a person's identity is critical to maintain subject-specific attributes. Existing methods either use adapter networks, which struggle to retain the facial details, structure & pose of the subject, or rely on full fine-tuning of large foundation models, which is computationally expensive and requires large high-quality annotated datasets. To overcome these limitations, we propose a novel unsupervised dataset preparation pipeline that enables scalable dataset generation and a novel identity-preserving loss function that ensures fine-grained identity preservation in the generated images. Despite using a significantly lighter foundation model and fine-tuning only a fraction of its weights, our method achieves performance comparable to state-of-the-art methods. Furthermore, it has robust generalization to out-of-training prompts and generalizes to multi-person images despite training only on single-person images.

## 1 Introduction

Recent advances in large text-to-image diffusion models, such as DALL·E (Ramesh et al., 2021; 2022), Imagen (Saharia et al., 2022; Baldridge et al., 2024), and Stable Diffusion (Rombach et al., 2022; Podell et al., 2023; Esser et al., 2024), have transformed image generation by enabling the synthesis of highly detailed and photorealistic images using natural language text prompts. An active area of research in this domain is personalized image generation and editing, which adapts the generated outputs to specific subjects or styles based on input text prompts. It has applications in AI-driven content creation, virtual advertising and social media content generation.

A critical challenge in personalized image editing is preserving fine-grained identity details (e.g., facial details and clothing) and full pose of foreground subjects during prompt-based image relighting. Identity preservation is highly sensitive to lighting direction and colour, which can affect the perceived identity (Adini et al., 1997; Braje et al., 1998; Varkarakis et al., 2021). Manipulating illumination while retaining fine-grained identity is an extremely difficult task.

We address this challenge across two distinct scenarios: 1) studio photography effects, where text prompts guide the model to generate window grill shadows and pantone effects (Fig. 1b); and 2) virtual backgrounds, where the subject is integrated into a scene generated from text prompts, with lighting of the scene faithfully reflected on the subject (Fig. 1b).

Existing methods for prompt-based personalized image editing use either inference-time training with few sample images (Ruiz et al., 2023; Gal et al., 2022; Parihar et al., 2024; Shi et al., 2024a) or adapter networks designed to provide greater control over style and identity (Zhang et al., 2023; Mou et al., 2024; Ye et al., 2023). Some approaches combine multiple adapters with multiple reference images (Wang et al., 2024; Li et al., 2024). However, these methods fail to capture fine-grained identity and pose, and largely fail on multi-person images. Zhang et al. (2025) fine-tune the full foundation model for robust identity preservation, but struggle to generate dramatic facial shadows.

We propose a novel method for prompt-based image relighting addressing these limitations. We design a novel unsupervised dataset preparation pipeline that generates diverse pairwise training images from only a few input samples. Additionally, we propose a novel identity loss, applied along with content losses, to ensure accurate identity preservation even when trained on imperfect training

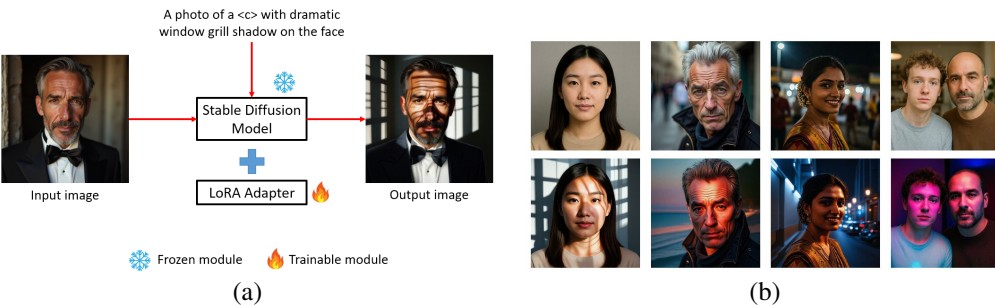

Figure 1: (a) Overview of the architecture for prompt-based image relighting. (b) Generated results from our model for four different prompts. Top row: Input images. Bottom row: Generated images.

data. We train a lightweight Low-Rank Adaptation (LoRA) (Hu et al., 2021) adapter that controls the subject identity in the generated image (Fig. 1a). Our method achieves comparable performance to state-of-the-art (SOTA) models despite using a much lighter foundation model and training only a fraction of parameters. Our contributions are:

- An unsupervised dataset preparation pipeline that generates large-scale pairwise training images from limited input image samples
- Identity and content losses that enable accurate learning of identity and facial shadows, even when trained on an imperfect dataset
- A training strategy that enables generalization to multi-person images despite training only on single-person images (Fig. 1b).

## 2 RELATED WORK

### 2.1 TEXT-TO-IMAGE GENERATION

Large text-to-image diffusion models (Ramesh et al., 2021; 2022; Rombach et al., 2022; Saharia et al., 2022; Nichol et al., 2021; Baldridge et al., 2024) generate high-quality, photorealistic images by training on large-scale datasets of captioned images (Schuhmann et al., 2022). These models are scaled to higher resolutions using two distinct approaches: 1) cascaded diffusion (Ramesh et al., 2022; 2021; Saharia et al., 2022), where a low-resolution image is generated and then upscaled to higher resolutions or 2) latent space training (Rombach et al., 2022), where diffusion models are trained in the compressed latent space, significantly reducing computational costs. Recent methods replace UNet backbone with Diffusion Transformers (DiT) Peebles & Xie (2023) to improve image fidelity and prompt adherence (Labs, 2024; Chen et al., 2023; Esser et al., 2024).

### 2.2 IMAGE EDITING

Image editing methods have evolved from latent space manipulations in GANs (Shen et al., 2020; Patashnik et al., 2021; Gal et al., 2022) to diffusion-based methods that offer greater flexibility. Some methods achieve image-to-image translation via iterative denoising (Meng et al., 2021; Lugmayr et al., 2022), while others control editing through cross-attentions (Hertz et al., 2022; Mokady et al., 2023). Recent methods explore fine-tuning the full diffusion model on image-prompt pairs (Brooks et al., 2023; Zhang et al., 2025; Kawar et al., 2023) and plug-and-play adapter networks (Zhang et al., 2023; Mou et al., 2024; Ye et al., 2023) using edges, segmentation, and pose maps. Some methods allow point-based interactive control for geometric manipulations (Pan et al., 2023; Shi et al., 2024b). Most prior work struggle to preserve the foreground identity and generate images having dramatic shadows on the face.

### 2.3 PERSONALIZATION AND IDENTITY PRESERVATION

Personalization techniques adapt the outputs of diffusion models to user-specific identity, style, or effects. Some methods fine-tune the model at inference on a few images (Ruiz et al., 2023) or embed

subject-specific tokens to control identity (Gal et al., 2022; Kumari et al., 2023). Other approaches either train adapter networks (Zhang et al., 2023; Mou et al., 2024; Ye et al., 2023; Li et al., 2024; Ruiz et al., 2024) or use training-free methods (Wang et al., 2024; Chen et al., 2023) for personalized editing. However, most of these methods fail to preserve fine-grained identity and pose, and they do not generalize to multi-person images. Alternatively, some methods use background environment maps as inputs and focus on accurate shadow generation with identity preservation (Ren et al., 2024; Chaturvedi et al., 2025; Kocsis et al., 2024). In contrast, our method does not require any environment maps or inference time fine-tuning, and it generalizes well to multi-person images while preserving the foreground identity and pose.

## 3 NOVEL DATASET PIPELINE

Prompt-based image relighting requires input-target image pairs that fully preserve the foreground subject's identity and pose while introducing diverse, photorealistic relighting effects. Publicly available datasets lack such pairwise images, and manually capturing them is both expensive and time-consuming. Thus, we designed a novel unsupervised dataset generation pipeline using the InstantID (Wang et al., 2024) model, which we modified for improved image fidelity.

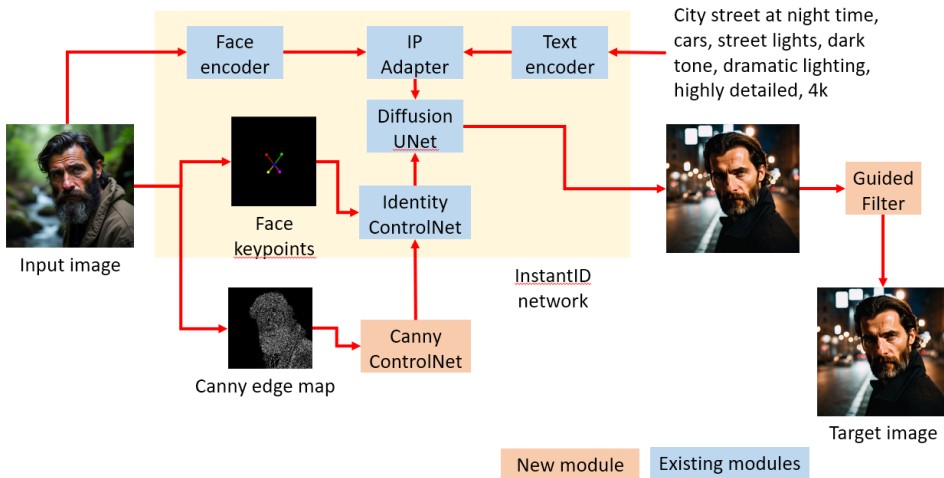

Figure 2: Architecture used for the unsupervised training dataset generation pipeline. We modified the original InstantID pipeline by adding a canny ControlNet.

Initially, we utilized InstantID's pipeline to generate the training dataset (indicated in yellow in Fig. 2). This model takes an input image and a text prompt as inputs and generates an edited output image based on the text prompt. While InstantID tries to preserve the identity, it often generates images with smoothed facial details, altered jawlines and face structures, and changes in the pose, clothing and orientation of the foreground subject (Fig. 3a). Additionally, the style, colour, and design of accessories such as glasses and jackets are significantly different in the generated image.

To mitigate these issues, we integrated a pre-trained ControlNet model (Zhang et al., 2023) into our pipeline (Fig. 2), which takes a dense canny edge map as input. This constraint significantly improves the pose and orientation of the foreground subject in the generated images (Fig. 3a). Additionally, we applied a wavelet transform-based guided filter (He et al., 2012) to refine fine-grained facial details, such as wrinkles and scars in the generated images (Fig. 3a). Although skin details improved, the generated images had inconsistencies in facial structure, expression, clothing, and accessories, leading to identity mismatches between the input and generated images.

However, InstantID largely maintained consistent identity across various prompts for a given input image (Fig. 3). Exploiting this property, we trained the model to learn mappings between InstantID generated images (Fig. 3c), rather than directly mapping the original input image to the InstantID generated image (Fig. 3b). For each original input image, we generated InstantID outputs for 67 diverse prompts (obtained from Zhang et al. (2025)). We created a pairwise dataset such that 60 of these outputs served as inputs and remaining 7 served as the ground truth (more details about the

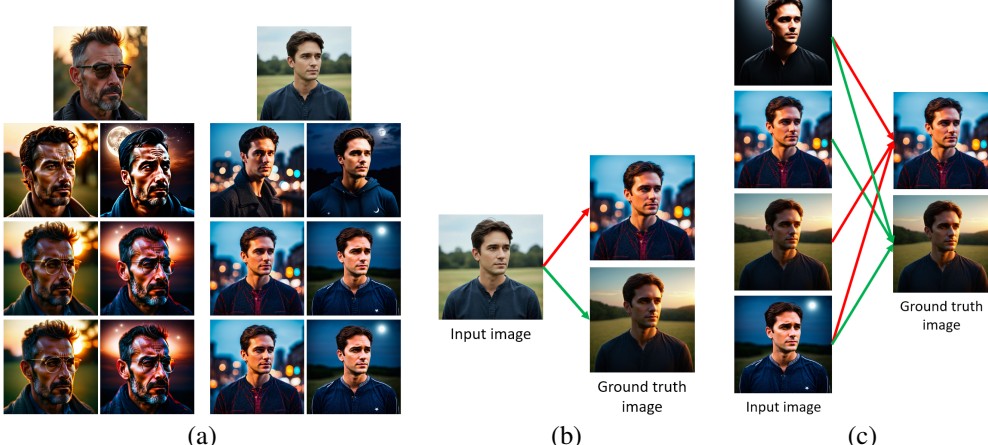

(a)                                    (b)                                    (c)

Figure 3: (a) Results from the dataset generation pipeline on two input images for two different prompts. First row: Original input image. Second row: Results from original InstantID pipeline. Third row: Results from our modified InstantID pipeline (Fig. 2). Fourth row: Results after applying guided filter on the images from the third row. (b) Standard approach trains the model to map the original input image to the target image. (c) Our proposed dataset generation strategy creates input-target training pairs from InstantID generated images. Each target image is paired with multiple generated images used as input (indicated with red and green arrows).

prompts are provided in appendix A). This approach ensured identity consistency between input-target pairs, thus enabling the model to learn the identity information. For each ground truth image, we generated 60 input images having diverse lighting conditions and variations, thus improving the generalization capabilities of the model. For each original input image, we created 420 input-target training pairs. Using only 1,024 single-person original input images from the publicly available SFHQ-T2I dataset (Beniaguev, 2024), our pipeline produced 430,080 training pairs, with 90% used for training and 10% for validation. This unsupervised framework enables efficient scaling of the training dataset by varying the number of supported prompts and original input images.

## 4 METHOD

### 4.1 PRELIMINARIES

**Foundation models** used for image generation can be broadly categorized into: 1) text-to-image (T2I) diffusion models and 2) image-to-image diffusion models. The former generates images conditioned only on a text prompt, while the latter generates images conditioned on both a text prompt and an input image. A widely used family of foundation models is Stable Diffusion model (Rombach et al., 2022), which applies diffusion in a compressed latent space. Training this model involves two steps: (a) training a Variational AutoEncoder (VAE) (Kingma & Welling, 2013) to encode the images into latent representations, and (b) training a diffusion model in this latent space, conditioned on the text prompt.

**Adapter networks** enable generalization of foundation models to downstream tasks by integrating modular networks without modifying the base foundation model. ControlNet (Zhang et al., 2023) duplicates the UNet for precise control. T2I adapter (Mou et al., 2024) downsamples the input image once and fuses it with the UNet at multiple scales. IP adapter combines text and image features through a decoupled cross-attention for flexible image and style editing (Ye et al., 2023). LoRA (Hu et al., 2021) learns low-rank matrices to update the model weights, enabling parameter-efficient fine-tuning with zero additional latency. These adapters support modular composition of models and preserve the base foundation model weights.

## 4.2 MODEL ARCHITECTURE

We use an image-to-image diffusion model with a LoRA adapter for prompt-based image relighting. Our approach adapts the text-to-image Koala foundation model (Lee et al., 2024) for image-to-image editing through a two-stage process: pre-training for image-conditioned generation, followed by fine-tuning for the relighting task.

The pre-training stage has two main goals: (1) adapting the text-to-image model for image-to-image editing, and (2) providing a good weight initialization for the fine-tuning stage. Inspired by the InstructPix2Pix model (Brooks et al., 2023), we concatenate the input latents with five additional channels and process the resulting 9-channel input through a trainable convolutional layer (*conv_in*). This layer projects the 9-channel input into the UNet feature space without re-training the full UNet. During pre-training, the model is optimized for input-to-input mapping, where the input image is passed to the model and it is trained to reconstruct the input image itself, conditioned on the prompt associated with the image. This enables the model to learn image-conditioned generation and ensures that the trained *conv_in* layer weights provide a good initialization for the fine-tuning stage.

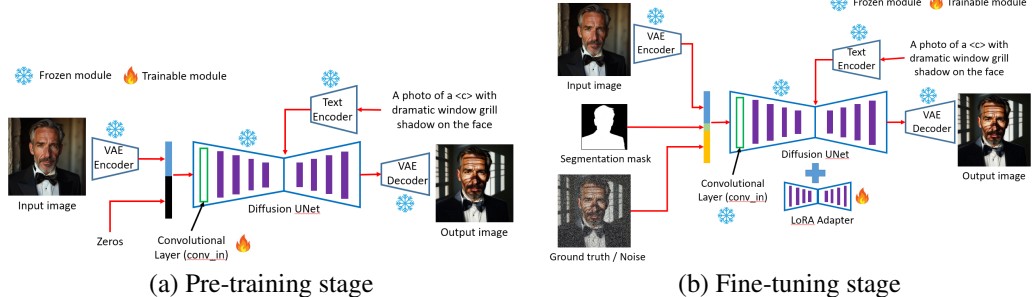

(a) Pre-training stage  (b) Fine-tuning stage

Figure 4: (a) Network architecture for pre-training stage, where the text-to-image model is adapted for image-to-image editing. Only the *conv_in* layer is trained at this stage. (b) Network architecture for fine-tuning the model for prompt-based image relighting. Only the LoRA adapter is trained at this stage.

The architecture used for pre-training is shown in Fig. 4a. The input image is first encoded into a 4-channel latent by the VAE encoder (blue in Fig. 4a) and then concatenated with five channels of zeros (black in Fig. 4a). The resulting 9-channel input is passed through the *conv_in* layer[1] (green in Fig. 4a), and then processed by the UNet, which predicts the noise at each timestep. The clean 4-channel latent is then estimated using the diffusion noise schedule and passed through the VAE decoder to reconstruct the input image. Only the *conv_in* layer is trained during this stage, while all other modules are frozen.

In the fine-tuning stage (Fig. 4b), the channels of zeros are replaced with a foreground segmentation mask and a 4-channel latent representing either the noised ground truth during training or the noised prediction during inference (green and orange in Fig. 4b). This 9-channel input is passed through the *conv_in* layer and the UNet to predict the noise at each timestep. Unlike pre-training, the model now estimates the clean 4-channel ground-truth (target) latent, which is reconstructed by the VAE decoder to the target image during training or the predicted image at inference. In this stage, only the LoRA adapters are trained on input–prompt–target triplets, while all other modules remain frozen.

We use a LoRA adapter for its efficiency and flexibility. It enables fast training as only a few low-rank matrices are updated, and it is agnostic to the base network architecture. It has negligible computational or storage cost at inference once the adapter weights are merged with the base model.

## 4.3 IDENTITY-PRESERVING PROMPT STRUCTURE

We use an identity-specific placeholder in the text prompts, following the prior work on image-to-image editing (Ruiz et al., 2023; Gal et al., 2022). The placeholder is selected such that it does not exist in the tokenizer's vocabulary, signaling to the model that it represents a special token. During

---

[1]This layer is integrated as the first layer of the UNet and is therefore shown as part of the UNet in Fig. 4.

training, we use the placeholder "a <c>" in the prompts, allowing the model to associate it with identity-specific visual features.

Most prior methods encode identity by fusing visual features into the text embeddings (Ruiz et al., 2023; Gal et al., 2022) or by using additional networks (Li et al., 2024; Wang et al., 2024; Ruiz et al., 2024). In contrast, our method uses only the placeholder prompt to encode the identity. During training, the novel loss functions guide the model to associate the placeholder token with the foreground subject's identity. As shown in Fig. 6a, using only the placeholder "a <c>" in the text prompt during inference generates images with accurate identity preservation. This indicates that the placeholder prompt effectively guides the model to learn and preserve identity-specific information.

To ensure the prompts remain semantically meaningful, we integrate the placeholder naturally with the descriptions of the relighting effects. For example, prompts for generating window grill shadow or a sunset lighting effects are: "a photo of a <c> with dramatic window grill shadow on the face" and "a <c> at a beautiful sunset at the beach, dramatic lighting, highly detailed, 4K".

## 4.4 TRAINING LOSSES

We used four losses to train the LoRA adapter: 1) identity loss, 2) foreground content loss, 3) background content loss and 4) noise loss.

Let $I \in \mathbb{R}^{H \times W \times 3}$ and $E(I) \in \mathbb{R}^{C \times H' \times W'}$ denote the input RGB image and its latent embedding output by the VAE encoder. Similarly, let $T \in \mathbb{R}^{H \times W \times 3}$ and $E(T) \in \mathbb{R}^{C \times H' \times W'}$ denote the target RGB image and its latent embedding output by the VAE encoder. Let $M \in [0,1]^{1 \times H' \times W'}$ denote the foreground segmentation mask of input image $I$, resized to the latent embedding resolution.

At diffusion timestep $t$, let $x_t$ be the noised VAE latent at time $t$, $\epsilon_\theta(x_t, t)$ be the UNet's predicted noise at time $t$ and $\beta_s$ be the noise scheduler. The cumulative noise coefficient, $\bar{\alpha}_t$, is defined as $\bar{\alpha}_t = \prod_{s=1}^{t}(1 - \beta_s)$. The denoised latent at timestep $t = 0$ is computed as

$$\hat{x}_0 \;=\; \frac{x_t \;-\; \sqrt{1 - \bar{\alpha}_t}\,\epsilon_\theta(x_t, t)}{\sqrt{\bar{\alpha}_t}} \tag{1}$$

**Identity loss.** We compute a novel identity loss ($\mathcal{L}_{\mathrm{id}}$) to ensure the foreground subject's identity is accurately preserved in the generated image.

$$\mathcal{L}_{\mathrm{id}} = 1 - CosSim\big(\tilde{x}_0,\ \tilde{x}_0^{(\mathrm{input})}\big) = 1 - \frac{\tilde{x}_0 \cdot \tilde{x}_0^{(\mathrm{input})}}{\|\tilde{x}_0\|_2 \|\tilde{x}_0^{(\mathrm{input})}\|_2} = 1 - \frac{(M \odot \hat{x}_0) \cdot (M \odot E(I))}{\|M \odot \hat{x}_0\|_2 \|M \odot E(I)\|_2} \tag{2}$$

where $CosSim$ computes the cosine similarity between the masked predicted image latents and masked input image latents. $\odot$ refers to element-wise multiplication.

Typically, losses are computed between the model's prediction and the ground truth. However, since we do not have paired training images with precise matching identity, we compute the identity loss between the input latent and the predicted latent. This encourages the model to preserve the foreground subject's identity in the generated image.

**Foreground content loss.** To accurately capture lighting details on the foreground subject, we compute a foreground content loss ($\mathcal{L}_{\mathrm{fg}}$) defined as the smooth L1 loss between the masked predicted image latents and ground truth latents ($\tilde{x}_0^{(\mathrm{target})} = M \odot E(T)$)

$$\mathcal{L}_{\mathrm{fg}} = \begin{cases} (\tilde{x}_0 - \tilde{x}_0^{(\mathrm{target})})^2/2 & \text{if } |\tilde{x}_0 - \tilde{x}_0^{(\mathrm{target})}| < 1 \\ |\tilde{x}_0 - \tilde{x}_0^{(\mathrm{target})}| - 0.5 & \text{otherwise} \end{cases} \tag{3}$$

**Background content loss.** We compute the background content loss ($\mathcal{L}_{\mathrm{bg}}$) to ensure that the generated backgrounds are accurate and photorealistic. It is defined as the L1 loss between the background regions of the predicted and ground truth latents.

$$\mathcal{L}_{\mathrm{bg}} = (1 - M) \odot \hat{x}_0 - (1 - M) \odot E(T) \tag{4}$$

Since the foundation model primarily contributes to background generation, this loss encourages the model to generate photorealistic outputs even when it lacks knowledge of photographic effects.

**Noise loss.** We compute noise loss ($\mathcal{L}_{\text{noise}}$) to guide the weight updates of the LoRA adapter during training. It is defined as

$$\mathcal{L}_{\text{noise}} = \mathbb{E}_{x_0, \epsilon, t}\left[\|\epsilon - \epsilon_\theta(x_t, t, c)\|_2^2\right] \tag{5}$$

where $x_t$ is the noised VAE latent obtained by adding Gaussian noise ($\epsilon \sim \mathcal{N}(0, I)$) corresponding to timestep $t$ to the clean image latent $x_0$. $c$ is conditioning context (text embeddings) and $\epsilon_\theta(x_t, t, c)$ is the predicted noise by the diffusion UNet.

The total loss used to train the LoRA adapter is

$$\mathcal{L}_{\text{total}} = \alpha_1 \mathcal{L}_{\text{id}} + \alpha_2 \mathcal{L}_{\text{fg}} + \alpha_3 \mathcal{L}_{\text{bg}} + \alpha_4 \mathcal{L}_{\text{noise}} \tag{6}$$

where $\alpha_1 = 1$, $\alpha_2 = 10$, $\alpha_3 = 0.2$ and $\alpha_4 = 0.5$ were determined empirically.

All losses are computed in the latent space rather than the image space, as this offers several advantages. The latent space is smoother and encodes rich semantic information, enabling the model to effectively learn both identity and lighting transformations while maintaining training stability. Additionally, computing losses in latent space improves computational efficiency by avoiding full image reconstruction at each training iteration, thereby reducing the impact of potential artifacts introduced by the VAE decoder.

### 4.5 TRAINING DETAILS

In the pre-training stage, the model is trained on $121,600$ image-caption pairs from the publicly available SFHQ dataset (Beniaguev, 2024). Given both the input image and its corresponding text prompt, the model is trained to reconstruct the input image itself using only the noise loss (Eq 5). As shown in Fig. 4a, only the *conv_in* layer is trained, while all other modules remain frozen. The initial learning rate of $1 \times 10^{-5}$ is decayed using a cosine annealing scheduler. The model is trained for $200,000$ iterations with a batch size of $8$, using the AdamW optimizer (Loshchilov & Hutter, 2017) with L2 regularization of $0.01$. The model was trained at a resolution of $1024 \times 1024$ pixels.

The model is fine-tuned on a training dataset generated using our novel unsupervised data generation pipeline described in Section 3. As shown in Fig. 4b, only the LoRA adapter was trained, while the remaining modules are frozen. We attach a rank 32 LoRA adapter to all convolution and attention layers of the UNet and train only these adapters while keeping the UNet layers frozen. The training loss is optimized using the AdamW optimizer with L2 regularization of $0.01$. The initial learning rate of $8 \times 10^{-5}$ is decayed using a cosine annealing scheduler. The model is trained for $100,000$ iterations with a batch size of $8$. Fine-tuning is also performed at a resolution of $1024 \times 1024$ pixels. Foreground segmentation masks are estimated using a pre-trained Mask R-CNN model (He et al., 2017) and resized to $128 \times 128$ pixels to segment the foreground regions in the image latents. All training and inference are performed on a single H100 GPU.

## 5 RESULTS

We evaluate our model on a challenging test dataset of 550 images, consisting of 400 images from the SFHQ-T2I dataset (Beniaguev, 2024) and 150 images from the celeb-FFHQ dataset (Karras et al., 2020). All test images were different from those used in training and validation. Both datasets consisted of single-person upper-body or face images generated using various T2I models. The test dataset covered a wide range of variations such as ethnicity, age groups, facial features (hair, structures), clothing styles and facial accessories (ear rings, caps, glasses). Each model was evaluated across 15 prompts, seven of which were used for training. Full details of the prompts are provided in appendix A.

The model performance was evaluated using three metrics: (a) CLIP score, which measures the semantic alignment between the text prompt and generated image; (b) Face ID score, which measures the dissimilarity in the face region between the input and generated images, and (c) Clothing score, which measures the dissimilarity in the clothing (non-face) regions between the input and generated images. More details about the metrics are provided in appendix B.

We compare the performance of our model against two SOTA prompt-based image relighting methods: InstantID (Wang et al., 2024) and IC Light (Zhang et al., 2025). To ensure a fair comparison, we

| | Total # parameters (in Billions) | # trainable parameters (in Billions) | Inference time (in seconds) | CLIP score ↑ | Face ID score ↓ | Clothing score ↓ |
|---|---|---|---|---|---|---|
| InstantID | 6.30 | 1.67 | 10.3 | 0.29 ± 0.03 | 0.31 ± 0.03 | 0.33 ± 0.02 |
| IC Light | 12.0 | 11.42 | 18.1 | **0.31 ± 0.02** | **0.20 ± 0.02** | 0.11 ± 0.03 |
| **Proposed** | **1.47** | **0.043** | **6.3** | 0.29 ± 0.02 | 0.20 ± 0.03 | **0.11 ± 0.02** |

Table 1: Quantitative comparison of our model with prior work on our test dataset.

exclude methods that require fine-tuning at inference (Ruiz et al., 2023; 2024; Parihar et al., 2024), rely on multiple reference images to learn the identity (Li et al., 2024) or use environment maps as additional inputs (Ren et al., 2024; Chaturvedi et al., 2025; Kocsis et al., 2024). As shown in Table 1, our model significantly outperforms InstantID, which used both ControlNet and IP-Adapter for identity preservation. Our method is comparable to IC Light, which fully fine-tunes the flux foundation model for image relighting. In contrast, our method uses a significantly smaller foundation model and trains only a lightweight LoRA adapter (43 million parameters, $\sim 4\%$ of the diffusion UNet). The CLIP score of our model is slightly lower than IC Light because the background is primarily generated by the foundation model and we use a much lighter foundation model.

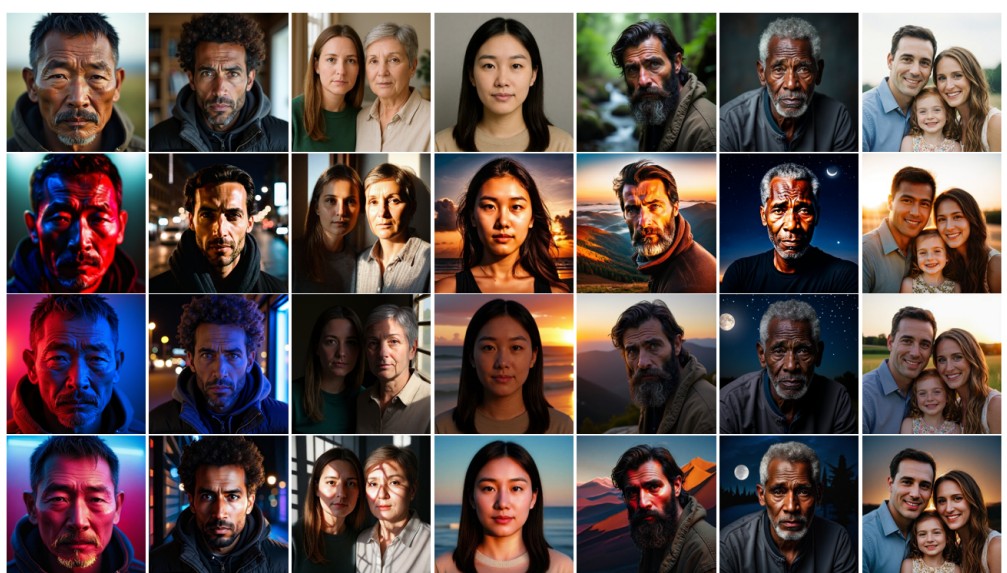

Figure 5: Qualitative comparison against prior methods on single-person and multi-person images. Row-1: Input image; Row-2: InstantID (Wang et al., 2024); Row-3: IC-Light (Zhang et al., 2025); Row-4: Proposed model. Columns are different effects (left to right): pantone, neon bokeh, window grill shadow, ocean sunset, mountain sunset, night sky, golden hour. The first three columns are part of the training prompts, while the remaining are out-of-training prompts.

Qualitative results in Fig. 5 further support the quantitative results in Table 1. Our model preserves identity significantly better than InstantID and generates images with noticeably more dramatic shadows than IC Light, while still preserving fine-grained identity details. Our results maintain the pose, orientation, expression, clothing style, and accessories of the foreground subject across diverse prompts, even though the model was trained on imperfect training data[2]. We attribute this to the identity loss, $\mathcal{L}_{\text{id}}$, which ensures faithful reconstruction of the identity-specific features in the generated images. More qualitative results are provided in the appendix C.

InstantID is limited to editing only the largest detected face in an input image, making it unsuitable for multi-person images. To enable a fair comparison on multi-person images, we use the modified canny ControlNet pipeline (Fig. 2) to generate the predicted images for InstantID[3]. Recall that, this

---

[2]The model is trained on image pairs generated using the modified InstantID pipeline, where the foreground subject's identity is not identical between the input and target images.

[3]Since this pipeline differs from the architecture of InstantID (Wang et al., 2024), we only compare the qualitative results.

pipeline was used to create our training dataset. As shown in Fig. 5, our model achieves better identity and pose preservation on multi-person images than the outputs from the modified InstantID pipeline. It also generates more dramatic facial shadows compared to IC Light. Notably, even though our training dataset was generated using the modified InstantID pipeline Fig. 2, our model generates more realistic images with improved identity preservation. This highlights the effectiveness of our loss functions and training strategy. Furthermore, despite being trained only on single-person images, our model generalizes effectively to multi-person images, indicating that the LoRA adapter has learned identity-preserving representations.

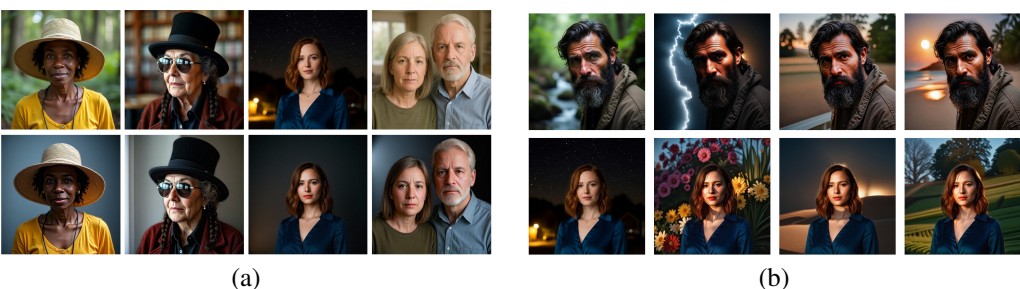

(a)                                                           (b)

Figure 6: (a) Results from our model using the placeholder prompt at inference. The foreground subject identity is accurately preserved in the generated image, while the background is random since no cue is given in the prompt. (b) Generalization of our model to out-of-training prompts. The foreground identity is similar to the input image (first column) and consistent across prompts.

To further evaluate the identity-preserving capabilities of the LoRA adapter, we infer the model using only the placeholder prompt "a <c>" along with the input image. As shown in Fig. 6a, the model accurately preserves fine-grained foreground identity details, indicating that the placeholder token effectively encodes identity features. Unlike prior methods that embed visual features into text embeddings (Ruiz et al., 2023; 2024; Li et al., 2024; Wang et al., 2024), our method does not require any additional visual features for identity preservation. Moreover, our model also generalizes well to out-of-training prompts, as shown in Fig. 6b. It preserves the foreground identity while generating appropriate lighting effects. This suggests that identity is primarily learned by the LoRA adapter, while the foundation model handles background generation and prompt-based variations.

## 6 LIMITATIONS AND FUTURE WORK

A limitation of our current method is the lack of fine-grained control over lighting parameters such as position and direction via the text prompt. While the foundation model can generate backgrounds with plausible lighting, our method does not support prompt-conditioned control of the lighting parameters across diverse variations. A promising future direction is to incorporate explicit prompt-based control over the lighting parameters, allowing for more precise control of the illumination and shadows. Another extension is to disentangle identity and lighting representations, enabling independent control of the identity and photorealism in the generated images.

## 7 CONCLUSION

We presented a novel system for fine-grained identity preservation in prompt-based image relighting. We proposed a novel unsupervised dataset generation pipeline that creates a large-scale pairwise training dataset from a few input images. Our approach adapts the Koala foundation model for image-to-image editing and trains a lightweight LoRA adapter with novel identity and content losses. We use a placeholder prompt "a <c>" to encode the identity information, and show that these tokens are sufficient for accurate identity preservation. Both qualitative and quantitative results show that our method achieve performance comparable with prior work, despite being trained on a significantly smaller foundation model and updating only a small fraction of its parameters. Furthermore, our model generalizes to multi-person images and out-of-training prompts. The proposed framework can be used to adapt other text-to-image foundation models for identity-preserving prompt-based image relighting.

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

## A  PROMPT DETAILS

As discussed in Section 5 of the main paper, we evaluated the model on 15 prompts, out of which only 7 were used for training the model. We evaluated the model on 8 out-of-training prompts to assess the generalization capabilities. The lighting effects and their corresponding prompts are listed below. The first 7 prompts were used for training the model.

1. window grill shadow : a photo of a  <c> with dramatic window grill shadow on the face

2. pantone : a colorful portrait of a  <c> in red and blue light, dramatic lighting

3. neon bokeh : cinematic bokeh a  <c>, bright city lights at night, film, bokeh, professional, 4k, highly detailed

4. city night : a portrait of a  <c> in middle of a city, night time, street, cars, dark tone, dramatic lighting, highly detailed, 4k

5. sunset silhouettes :  a  <c> outdoors during sunset, with the face partially silhouetted against the warm orange and pink hues, creating soft but visible shadows

6. dark shadow : a dark portrait of a  <c>, vantablack background, highly detailed, 4K

7. rainy day : a  <c> in a rainy city night, wet rain drops on a  <c>, highly detailed, 4k

8. ocean sunset : a  <c> at a beautiful sunset at the beach, dramatic lighting, highly detailed, 4K

9. mountain sunset : a cinematic portrait of a  <c> in backdrop of colourful mountains at dramatic sunset, sunlight falling on the face of a  <c>, dramatic lighting, amazing sky, studio, 4k

10. golden hour : a cinematic portrait of a  <c> at golden hour, light falling on the face of a  <c>, dramatic shadows, amazing sky, studio, 4k

11. night sky : a cinematic portrait of a  <c> in the backdrop of a dramatic night sky with stars and moon, dramatic shadow on the face, super realistic, highly detailed, 4k

12. garden : a cinematic portrait of a  <c> in a garden with colourful flowers, golden hour, dramatic shadow on the face, highly realistic, 4k

13. rural farm : a portrait of a  <c> at dusk in rural farm, with soft shadows and a serene natural setting, 4k, super realistic, high resolution

14. lightning flash : a portrait of a  <c> outdoors with a sudden flash of lightning casting stark, dramatic shadows and illuminating the face

15. desert sunrise : a portrait of a  <c>, witnessing a desert sunset with a mirage-like glow, surrounded by intricate sandy textures and warm tones, 4k, super realistic, high resolution

## B  EVALUATION METRICS

We evaluated the performance of our model using three metrics:

1. CLIP score: Measures the semantic alignment between the text prompt and the generated image. Let $I$ be the input image and $T$ be the text prompt. Let $e_I$ and $e_T$ be the image embeddings and text embeddings obtained using the CLIP image encoder and CLIP text encoder (Radford et al., 2021). The CLIP score is computed as the cosine similarity between these embeddings

$$\text{CLIP score} = \frac{e_I \cdot e_T}{\|e_I\| \, \|e_T\|} \tag{7}$$

2. Face ID score: Measures the cosine dissimilarity between the face regions of the input image and generated image. The face region is detected using the Retina Face model (Deng et al., 2019) and face features are extracted using FaceNet model (Schroff et al., 2015). Let $f_I$ and $f_P$ be the embeddings from FaceNet for the face regions in the input and generated images, respectively. The face ID score is computed as

$$\text{Face ID score} = 1 - \frac{f_I \cdot f_P}{\|f_I\| \, \|f_P\|} \tag{8}$$

3. Clothing score: Measures the cosine dissimilarity between the clothing (non-face) regions of the input image and generated image. The face region is subtracted from the foreground segmentation mask and the resulting mask is passed to the VGG network (Simonyan & Zisserman, 2014) to extract the image features. Let $c_I$ and $c_P$ be the embeddings from VGG network for the clothing regions in the input and generated images, respectively. The clothing score is computed as

$$\text{Clothing score} = 1 - \frac{c_I \cdot c_P}{\|c_I\| \, \|c_P\|} \tag{9}$$

## C  RESULTS

Fig. 7 shows some more qualitative comparison of our model against two prior works: InstantID (Wang et al., 2024) and IC-Light (Zhang et al., 2025). Our model preserves the foreground subject's identity and pose significantly better than InstantID and achieves more dramatic facial shadows than IC Light, while fully preserving the foreground identity. Additionally, our model generalizes effectively across different genders, ethnicities, facial structures, poses, and facial accessories.

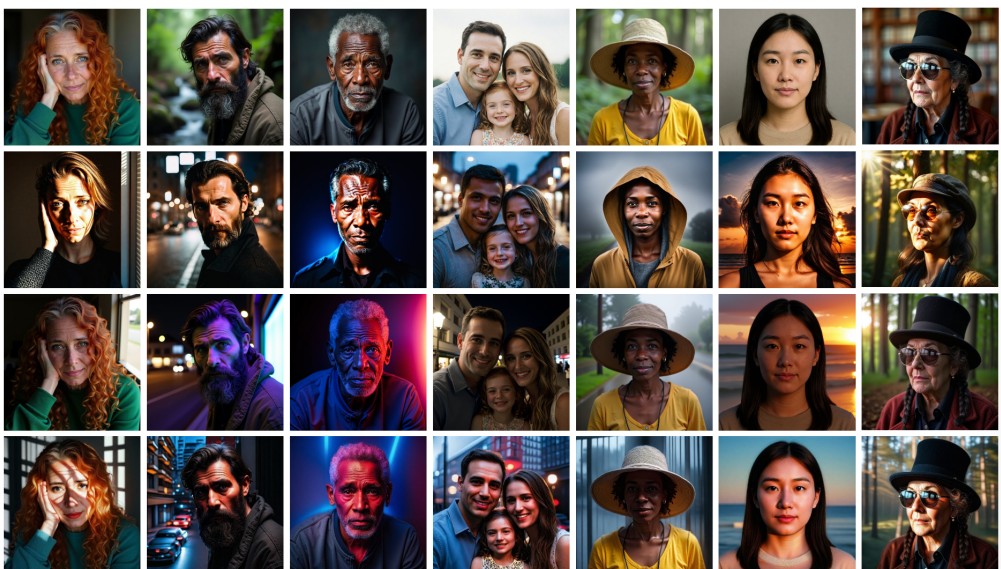

Figure 7: Qualitative comparison against prior methods on different test images. Row-1: Input image; Row-2: InstantID (Wang et al., 2024) results; Row-3: IC-Light (Zhang et al., 2025) results; Row-4: Proposed model. Left to right are different effects: window grill, city night, pantone, city night, rainy day, ocean sunset, forest canopy. The last two are out-of-training prompts, while the rest are used in training the model.

Fig. 8 shows the outputs generated by our model for different prompts on the same input image. The foreground subject's identity is consistent across different prompts and closely matches with the identity in input image, thus showcasing the generalization capabilities of our model across prompts with diverse lighting and backgrounds.

As described in Section 3 of the main paper, we propose a novel scalable dataset generation pipeline to create pairwise training images. Since InstantID fails to preserve the foreground subject's identity

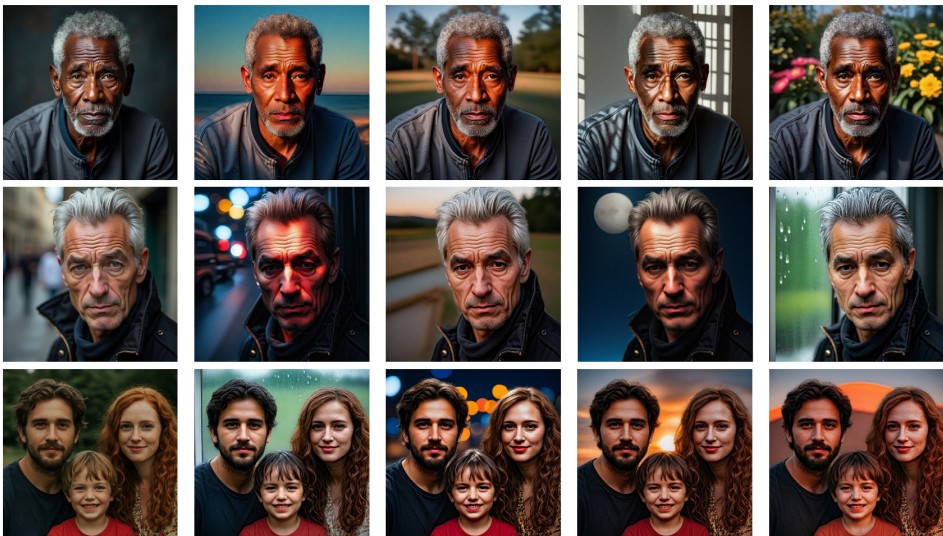

Figure 8: Generalization of our model across different prompts for the same input image. The foreground identity is consistent across prompts and similar to the input image. First column: Input image. Other columns: Generated results.

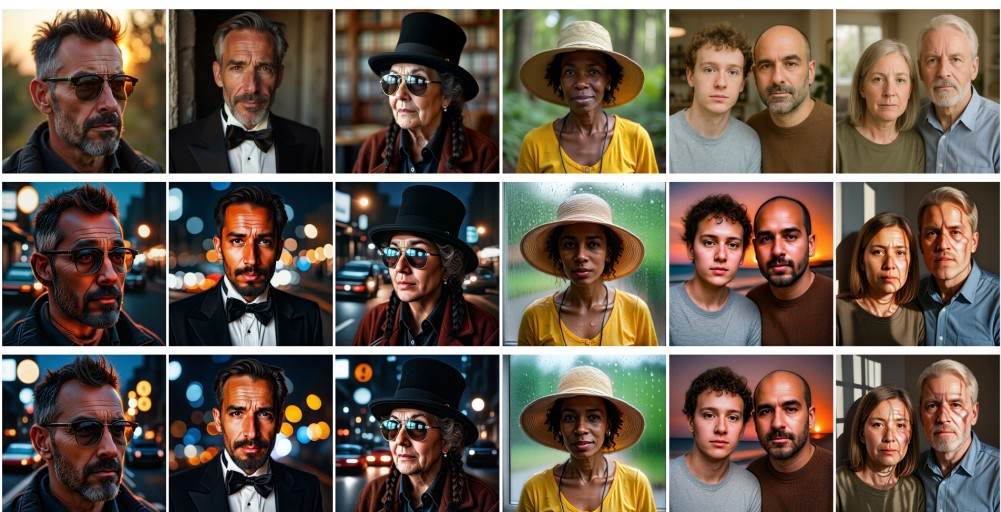

Figure 9: Qualitative comparison of the model trained with and without the pairwise dataset strategy. Top row: Input image. Middle row: Model trained directly on original input image and target pairs. Bottom row: Model trained on the pairwise dataset composed of input-target pairs both generated by InstantID.

and pose accurately, we created training pairs where the input and target images were generated by InstantID (see Fig 3c in the main paper). To evaluate the benefits of this pairwise dataset creation strategy, we compare a baseline model trained on original input-target image pairs with our model trained using the pairwise dataset strategy. As shown in Fig. 9, the identity preservation in the baseline model is significantly worse than our model trained using our pairwise dataset strategy.

As described in the main paper, we use the placeholder prompt "a <c>" during training to encode the identity information. Fig. 10 shows some more qualitative results generated using the placeholder prompt along with the input image at inference. The model fully preserves the pose and identity of the foreground subject, indicating that the placeholder prompt mainly learns the identity details, while the foundation model generates the background content.

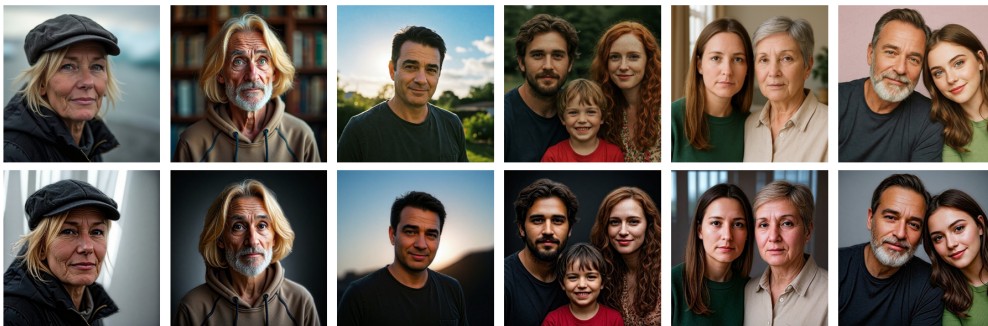

Figure 10: Results from our model using only the placeholder prompt at inference. The foreground subject identity is accurately represented in the generated image. The background is generated randomly, since no cue is passed into the prompt. Top row: Input images. Bottom row: Generated results.

To evaluate the effectiveness of our design choices, we performed several ablation studies. A key benefit of our unsupervised dataset generation pipeline is its ability to efficiently scale the size of training dataset using only a few original image samples. We evaluated the impact of varying the number of original images used to generate the training dataset. Specifically, we randomly sample subsets of the original input images and apply our novel dataset generation pipeline to create the training dataset. As shown in Table 2, performance improves slightly with larger datasets but quickly saturates as both 512 and 1024 samples achieve comparable results. This indicates that our pipeline generates training data with sufficient diversity and variation, even from limited number of original input images. All models were trained using the same hyperparameter configuration.

| # original samples | # training samples | CLIP score $\uparrow$ | Face ID score $\downarrow$ | Clothing score $\downarrow$ |
|---|---|---|---|---|
| 128 | 53,760 | 0.21 | 0.25 | 0.17 |
| 256 | 107,520 | 0.24 | 0.24 | 0.15 |
| 512 | 215,040 | 0.26 | 0.22 | 0.12 |
| 1024 | 430,080 | 0.29 | 0.20 | 0.11 |

Table 2: Performance of the model when trained on a dataset generated using fewer original input images samples.

Next, we evaluate the impact of LoRA adapter rank on the model performance. Adapter with higher ranks have more trainable parameters, which should enable the model to learn better identity and lighting details. However, as shown in Table 3, performance improvement with higher ranks are minimal. The performance of rank 32 LoRA is comparable with rank 64 and rank 128. The face ID score and clothing score degrade at lower ranks and stabilize at higher ranks, indicating that the LoRA adapter primarily contributes to learning foreground identity. The minimal benefits from higher rank adapters highlights the benefits of our novel loss functions and training methodology.

| Rank | # parameters (in Millions) | CLIP score $\uparrow$ | Face ID score $\downarrow$ | Clothing score $\downarrow$ |
|---|---|---|---|---|
| 16 | 21.7 | 0.26 | 0.30 | 0.18 |
| 32 | 43.4 | 0.29 | 0.20 | 0.11 |
| 64 | 86.9 | 0.29 | 0.20 | 0.11 |
| 128 | 173.7 | 0.30 | 0.19 | 0.11 |

Table 3: Performance comparison of our model for different ranks of the LoRA adapter.

Finally, we evaluate the contribution of each loss component in the training loss function (see Eq 6 in the main paper). The weight of each component is set to 0, while keeping the remaining weights unchanged. As shown in Table 4, setting $\alpha_1 = 0$ reduces the accuracy of foreground identity preservation, as reflected in a higher face ID score. Setting $\alpha_2 = 0$ affects the dramatic lighting and

shadow on the foreground, resulting in a lower CLIP score. Setting $\alpha_3 = 0$ reduces the photorealism of the generated backgrounds. The best performance is achieved when all components of the loss function are optimally used.

| Loss weight | CLIP score $\uparrow$ | Face ID score $\downarrow$ | Clothing score $\downarrow$ |
|---|---|---|---|
| $\alpha_1 = 0$ | 0.26 | 0.26 | 0.11 |
| $\alpha_2 = 0$ | 0.23 | 0.21 | 0.11 |
| $\alpha_3 = 0$ | 0.24 | 0.22 | 0.13 |

Table 4: Performance comparison of the contribution of different components of the training loss function.