# OpenReview forum: "IDetail: Fine-grained Identity Preservation in Prompt-based Image Relighting"
_ICLR.cc/2026/Conference — ICLR 2026 Conference Withdrawn Submission_

### Official Review · Reviewer_Dfyp · 2025-10-26

**Soundness:** 2
**Presentation:** 2
**Contribution:** 2
**Rating:** 2
**Confidence:** 4

**Summary:**

This work addresses the prompt-based image relighting problem in the field of personalized editing by proposing a data generation pipeline based on InstantID, enabling scalable dataset generation. It introduces a two-stage LoRA training approach and diverse training losses to ensure fine-grained identity preservation in the generated images, achieving competitive results.

**Strengths:**

- The approach uses a smaller base model and trains fewer parameters, yet achieves competitive results compared to Flux-based methods.
- It introduces a pipeline for automatic data generation, eliminating the need for manual data collection and annotation.

**Weaknesses:**

- The absence of ablation experiments makes it difficult to assess whether each module's design is essential, which is a critical flaw.
- Is the training setup for the first stage reasonable? It’s evident that the first stage training forces the model to ignore the newly added five input channels as much as possible, which doesn't provide a good initialization for the second stage.
- The design of multiple losses intuitively seems arbitrary and lacks clear justification.
- The method is not particularly novel, from data collection to model architecture design.
- Since the datasets are entirely generated by InstantID, they inevitably inherit any biases present in InstantID. As a result, models trained on these datasets will also inherit such biases. For instance, if InstantID performs poorly with certain prompts, this method will also struggle in those cases.
- The paper is not well-written, with many awkward sentences, and some words in Figure 2 and Figure 4 still have red error underlines beneath them.

**Questions:**

- Is the ability to generalize to multiple people a unique advantage of your method, or can other methods also achieve this?
- Will the results still be good when the character is farther from the camera?

---

### Official Review · Reviewer_87ka · 2025-10-30

**Soundness:** 1
**Presentation:** 2
**Contribution:** 1
**Rating:** 2
**Confidence:** 4

**Summary:**

This paper tackles prompt-based image relighting for human subjects, with a strong focus on preserving fine-grained identity under dramatic lighting edits.
The authors argue that existing approaches fail to retain the facial details, structure & pose of the subject and are computationally expensive to finetune the large models.

To address the problem of prompt-based person image relighting, this paper presents the following contributions:

* An unsupervised data generation strategy to produce the training (input image, prompts, relighted image) pairs.
* A two stage training procedure to adapt text-to-image diffustion model for image-to-image mapping.
* A combination of diffsuion loss with face identity loss, foreground/background content loss to finetune the diffusion loss.

Experimental results compared with personalized image generation method InstantID and prompt-based image relighting method IC-Light reflect the proposed model achieves a worse performance with a significantly smaller backbone.

**Strengths:**

1. The paper is well structured.
2. The proposed pipeline is more efficient than SOTA prompt-based image relighting method.

**Weaknesses:**

1. The rationality of using InstantID to generate relighted images is not justified. The relighted images in the paper looks bad.
2. The base diffusion model Koala from Neurips 2024 used in this paper is too old. There exists some stronger image-to-image generative models like Flux.1 Kontext. Why using such a weak model is not justified.
3. There is no ablation study in the paper.
4. The evaluation looks not rational in the paper. There are no any metrics related to relighting quality.
5. The qualitative results look significantly worse than the IC-Light. It doesn't make sense to call it comparable.
6. The face identity loss is computed in the latent space of VAE, which is different from the common practice of computing loss in the feature space of face recognition network. There is no justification for this design.

**Questions:**

Please see the weakness part.

---

### Official Review · Reviewer_RmV4 · 2025-10-31

**Soundness:** 1
**Presentation:** 2
**Contribution:** 1
**Rating:** 0
**Confidence:** 4

**Summary:**

This system introduce prompt-based image relighting framework upon denoising diffusion models.

**Strengths:**

- Prompt-based approach: For relighting, this system only uses prompt and input image, providing flexible and easy controls.
- Lighting weight manner: In Tab.1, they reported the system has significantly less parameters and relatively fast inference time.

**Weaknesses:**

- Lack of fidelity: There is no explicit relighting-specialized module, it is thereby observed the proposed system has fidelity concerns. Specifically, even though this task aims to only change the lighting-relevant factors, the resulting outcomes showed changed background appearance and color hair. This tendency dilute the goal of relighting, it rather seems like image editing task with high-level (fine) color changes.
- Lack of quantitative comparison: Experiments report the slight improvement in clothing score, and lower CLIP and ID score. There is no relighting measurement metric. It is strongly recommended to provide more comprehensive comparison to validate the proposed system in terms of relighting.
- Combination of existing module: The system consists of existing approaches including LoRA and loss functions with no novel learning algorithm.

**Questions:**

- It is strongly recommended to provide more comprehensive experiments to validate the capabilities of the proposed system.
- The authors should discuss why the results showed background changes despite the relighting task.

---

### Official Review · Reviewer_qwnC · 2025-10-31

**Soundness:** 2
**Presentation:** 2
**Contribution:** 2
**Rating:** 4
**Confidence:** 4

**Summary:**

This paper addresses the challenge image editing: preserving fine-grained identity details during prompt-based relighting. The proposed  **IDETAIL** is a prompt-based relighting method that preserves identity through a novel dataset generation pipeline and specialized loss functions, using lightweight LoRA adapters instead of full model fine-tuning.

**Strengths:**

1. The unsupervised approach generating 430K pairs from limited data addresses the scarcity of paired relighting data.
2. Achieving competitive results while training only LoRA adapters (43M parameters) compared to full model fine-tuning approaches.

**Weaknesses:**

1. The authors explicitly state that InstantID generates images with "smoothed facial details, altered jawlines and face structures, and changes in pose, clothing and orientation" (Section 3, Fig. 3a). Yet, they use these imperfect InstantID outputs as training targets. How can the model learn "fine-grained identity preservation" from data that doesn't preserve identity accurately? The entire training paradigm relies on the assumption that learning mappings between InstantID outputs will somehow yield better identity preservation than InstantID itself. This circular logic isn't sufficiently justified.
2. The SOTA methods move towards 3D understanding for relighting, such as incorporating monocular depth estimation, surface normal prediction, to understand how light interacts with surfaces in a physically plausible way. This approach is fundamentally 2D. It relies on the dataset pipeline and loss functions to implicitly learn lighting effects. The foreground segmentation mask provides a binary separation but no geometric information. This limits the model's ability to reason about physics. The "dramatic shadows" it generates, while visually appealing, may lack geometric consistency.

**Questions:**

1. How do you reconcile using identity-degraded InstantID outputs as training targets with your goal of fine-grained identity preservation? What specific aspects of your method overcome the limitations of the training data?
2. The experiment with the placeholder token "a <c>" (Fig. 6a/10) shows impressive identity preservation. However, this could be explained by the LoRA adapter learning to ignore an uninformative prompt and simply reconstruct the input latent. Can you provide further analysis (e.g., attention visualizations, feature space analysis) to prove the token is actively encoding identity features rather than just triggering a reconstruction mode?

---

### Author Response · Authors · 2025-11-18
**Clarifications and explanations for the reviewer comments**

We thank the reviewers for their feedback. Several points are already addressed in the paper. We give additional justification below.

1. Clarification on problem definition:
	Our task is prompt-based image relighting with two scenarios:
	1. Environmental relighting: the subject is transported to a new scene defined by the prompt, and lighting from that scene is realistically reflected on the subject. The generated image looks like the subject was photographed in the scene.
	2. Studio photography effects: prompts such as "pantone" or "window grill" simulate controlled studio captures where the backgrounds are simple to enhance the contrast with shadows and lighting.
	Both scenarios require relighting and background change.

2. Motivation for using InstantID training pairs:
	We require image pairs with: 1) precise identity preservation and 2) diverse background/lighting. Since no public dataset provides this, we generated training pairs using InstantID.
While the identity in InstantID generated images is inaccurate w.r.t original input image, the identity is consistent across different prompts. So we train the model on InstantID -> InstantID pairs.
	Our identity loss (Eq 2), computed between the predicted and input image latents (Section 4.4), corrects InstantID’s identity drift. This preserves identity at inference (Fig 5,7).

3. Two-stage training pipeline:
The base UNet expects 4-channel conv_in, but we use a 9-channel conv_in. Training the conv_in and LoRA simultaneously causes degradation in image quality. The two-stage pipeline, first adapts the conv_in to 9 channels while keeping generation quality (stage 1) and then trains only the LoRA for identity preservation (stage 2).

4. Loss design and latent space computation:
	Each loss targets a specific requirement:
	1. ID loss (Eq 2): identity preservation
	2. FG loss (Eq 3): preserves sharp lighting and shadows
	3. BG loss (Eq 4): backgrounds adhere to prompts
	4. Noise loss (Eq 5): for diffusion training

	Ablations in Table 4 (Appendix) show all losses are required for best performance. Losses are computed in the VAE latent space because it is semantically more meaningful and avoids decoder reconstruction artifacts (Section 4.4).

5. Ablations and model capabilities:
	Tables 2-4 (Appendix) show ablations on training dataset size, LoRA rank and loss component importance.
Fig 9 shows identity degradation when training on original input -> InstantID pairs, rather than InstantID -> InstantID pairs. Fig 5-10 shows generalization to multi-person images despite training only on single-person images, generalization to out-of-training prompts, importance of placeholder prompt "a <c>" for identity preservation.

6. Non-suitability of traditional relighting metrics:
	Relighting metrics such as PSNR or SSIM are computed at pixel-level and assume a single ground truth output, which is not applicable to our problem. Multiple valid outputs may exist for the same prompt (e.g., window grill shadow may vary in direction and pattern).
	We use metrics from generative image editing to evaluate two aspects: 1) identity preservation and 2) prompt adherence. Both face and clothing scores are computed to reduce bias. Prompt adherence is measured via CLIP score.

7. Not using more powerful foundation models:
	We use a lightweight foundation model so the benefit of our design choices are clearly attributable, which would be difficult with a larger backbone.
	Our model has 1.2B params with 43M trained. IC-Light has 12B params with 11.4B trained. Despite being 12x smaller and training 265x fewer parameters than IC-Light, we achieve comparable identity preservation and more dramatic lighting and shadows (Fig 5,7). This shows that improvements come from our design choices.

8. Qualitative comparison with IC-Light:
	Image resolution reduces the perceived quality of the results. Higher resolution comparisons are shown in Appendix. Our model preserves identity similar to IC-Light while generating more dramatic lighting and shadows.

9. Why 3D maps are not used:
	Although no explicit 3D geometry is used, the model generates physically plausible lighting (e.g., shadows bend along the face in window grill effect in Fig 5, 7). This is due to the strong priors learnt by pretrained foundation models. Using depth/normal maps would increase inference time with little benefit.

10. Additional clarifications: 1) InstantID dataset may transfer to our model, but results in Table 1 and Fig 5, 7 show clear improvement over InstantID due to our design choices. 2) Our model works on small faces (Fig 6b). There is no inherent limitation.

Several concerns and weaknesses were already addressed in the paper. We provide additional justifications above. We request the reviewers to reconsider their scores.

In the rebuttal revision, we will include: 1) results with SDXL foundation model, 2) more details on two-stage training, and 3) visualizations about the identity learnt by placeholder prompt.

---

### Note · Authors · 2025-11-29

**Comment:**

The reviewers missed several important details already mentioned in the paper. Given that the discussion phase is reverted, we are withdrawing the paper.

**Withdrawal Confirmation:**

I have read and agree with the venue's withdrawal policy on behalf of myself and my co-authors.